# A Novel LSTM and Graph Neural Networks Approach for Cardiovascular Simulations

Angelica Iacovelli
*Department of Pediatrics*
*Stanford University*
Stanford, CA 94305 USA
angy@stanford.edu

Luca Pegolotti
*Institute for Computational*
*and Mathematical Engineering*
*Stanford University*
Stanford, CA 94305 USA
lpego@stanford.edu

Matteo Salvador
*Institute for Computational*
*and Mathematical Engineering*
*Stanford University*
Stanford, CA 94305 USA
msalvad@stanford.edu

Edoardo Stoppa
*Department of Computer Science*
*Purdue University*
West Lafayette, IN 47907 USA
estoppa@purdue.edu

Marco D. Santambrogio
*Department of Electronics,*
*Information and Bioengineering*
*Politecnico di Milano*
Milano, MI 20133 Italy
marco.santambrogio@polimi.it

Alison Marsden
*Department of Bioengineering*
*Stanford University*
Stanford, CA 94305 USA
amarsden@stanford.edu

*Abstract*—We propose a novel method that integrates Long Short-Term Memory (LSTM) networks with Graph Neural Networks (GNNs) to build reduced-order models of cardiovascular simulations. Reduced-order models are often used as an alternative to full three-dimensional cardiovascular simulations, providing a way to simplify the computational demands associated with fully detailed 3D simulations. The proposed method encodes blood fluid dynamics within a MeshGraphNet-based framework, which is particularly effective in modeling complex physical systems by leveraging graph structures to represent the state of the system. Our method extends the capabilities of the original framework by incorporating LSTMs to capture long-term dependencies, thereby improving predictive accuracy and significantly reducing the computational resources required for the training process. This method achieves errors below 2% for blood pressure and flow rate predictions, showcasing a 65% improvement in average error rates compared to the MeshGraphNet-based framework and a notable increase in computational efficiency, reducing training time by at least 57%. Our method also introduces the ability to adapt the simulation to different cardiac cycles depending on the patient, providing a robust and efficient tool for patient-specific cardiovascular modeling.

*Index Terms*—Cardiovascular modeling, Graph Neural Networks, Long Short-term Memory Networks, Reduced-order modeling.

## I. Introduction

Cardiovascular disease remains the leading cause of premature death and disability in humans, and its incidence is on the rise globally, presenting an ongoing challenge for medical research and healthcare [1]. Over the past two decades, the field of Computational Fluid Dynamics (CFD) has played a pivotal role in cardiovascular research [2]–[4]. In CFD, the Navier-Stokes equations are numerically solved for patient-specific blood flow simulations, which enable personalized models and detailed analysis, such as evaluating wall shear stress, velocity, and pressure fields.

However, clinical deployment of cardiovascular simulations is severely limited by their excessive computational cost. Indeed, the simulation of a single heartbeat for an anatomically accurate patient-specific model can require several hours of computational time even on a supercomputer platform [5]. To address this, physics-based Reduced-Order Models (ROMs), simplifying the complexity of vessel geometry and reducing the variables needed to describe key quantities, have been developed [6]. ROMs include zero-dimensional and one-dimensional models [6]. Zero-dimensional models describe the cardiovascular system using an electrical circuit, where blood flow and pressure drops are analogous to electric currents and potential differences, respectively. These models do not depend on spatial variables. On the other hand, one-dimensional models simplify the three-dimensional Navier-Stokes equations to a single spatial dimension, representing arterial trees as segments and focusing on axial components of pressure, flow rate, and vessel wall displacement.

Both zero- and one-dimensional models often yield accurate results with a lower computational burden compared to full three-dimensional simulations. However, challenges arise, especially in accurately representing pressure losses at vascular junctions and modeling pathological cases like stenosis or aneurysms [7]. These limitations led to a growing interest in data-driven approaches such as Physics-Informed Neural Networks (PINNs) [8]–[10], Latent Neural Ordinary Differential Equations (LNODEs) [5], and Deep Operator Networks (DeepONets) [11]. These methods leverage the large amount of ground truth data generated from simulations, enabling the

The paper was submitted on June 12, 2024. This work was supported by the Ermenegildo Zegna Founder's Scholarship.

models to learn complex patterns and relationships within the data. Although these methods have the flexibility to be tailored to specific geometries using interpolation techniques [12]–[15], they often fall short of fully capturing the diverse and complex geometric variations that are typical in patient-specific anatomical models.

To overcome these difficulties, Graph Neural Networks (GNNs) have emerged as a promising alternative to traditional fully connected and convolutional neural networks. Although initially not used for patient-specific cardiovascular simulations, GNNs have shown adaptability to complex geometries. They have been effectively utilized in learning particle interaction laws [16], as solvers, such as MeshGraphNet, in mesh-based simulations [17], and have demonstrated their capabilities in predicting steady blood flow within three-dimensional arterial structures [18].

GNNs have also been recently utilized for learning ROMs for cardiovascular simulations [7]. In this MeshGraphNet-based method, the network iteratively considers the state of the system, comprising pressure and flow rate at a particular timestep and other relevant features, and computes approximations for the next values of pressure and flow rate. This model outperforms physics-driven one-dimensional models in geometries characterized by many junctions or pathological conditions. However, the accuracy of the model may be limited by its inability to account for long-term dependencies inherent in the cardiovascular system. Furthermore, there is the opportunity to enhance its generalization capabilities, particularly in integrating patient-specific features more effectively. This would allow for a more personalized approach to analyzing cardiovascular data.

To address these challenges, this work explores the integration of Long Short-Term Memory (LSTM) [19] networks in a GNN-based framework. LSTMs are particularly effective in tasks that require capturing long-term dependencies [20], such as in natural language processing and time-series analysis, but also in statistics, linguistics, medicine, and transportation [21]. While incorporating LSTMs into the proposed approach, the graph structure was retained to effectively model the geometries. This integration ensures that while harnessing the strengths of LSTMs for handling temporal dependencies, the spatial and structural complexities of cardiovascular geometries are still adequately represented through the graph-based framework. Indeed, Graph LSTMs can be used to enhance the efficiency of propagating long-term information across the graph structure [22].

Our contributions include a novel methodology to improve the MeshGraphNet-based method [7] performance. In summary, the main contributions of this article are:

1) Development and validation of a novel approach for cardiovascular simulations that integrates LSTM networks into a GNN framework. This resulted in several advantages:

   a) Accuracy Improvement: Our method exhibits a notable increase in accuracy, significantly reducing the average error compared to the MeshGraphNet-based method.

   b) GPU Training Efficiency: We achieved a significant improvement in training efficiency on GPUs, with our model completing epochs considerably faster than the MeshGraphNet-based method.

   c) CPU Training Efficiency: Similarly, our method shows enhanced efficiency in CPU training times per epoch, requiring substantially less time compared to the MeshGraphNet-based method.

2) Flexibility in Cardiac Cycle Period: Our method introduces the ability to select different periods for a cardiac cycle for each patient, offering improved adaptability and customization potential, contrasting the fixed period approach of the MeshGraphNet-based method, potentially leading to better generalizability in diverse simulation scenarios.

The paper is structured to first provide a comprehensive background, followed by a description of our method, results, and a thorough discussion of our findings and their implications for future cardiovascular research.

## II. BACKGROUND AND RELATED WORKS

In this section, we delve into the progression and current state of computational approaches in cardiovascular research. We start with physics-based one-dimensional ROMs and then explore advanced data-driven methods based on GNNs.

### A. Physics-based models for cardiovascular applications

Let us consider one-dimensional models designed to approximate the characteristics of compliant blood vessels, focusing on pressure, flow rate, and wall displacement along the central axis. These models are derived from the three-dimensional Navier-Stokes equations, given by:

$$\nabla \cdot \mathbf{u} = 0, \tag{1}$$

$$\frac{\partial \mathbf{u}}{\partial t} + (\mathbf{u} \cdot \nabla)\mathbf{u} = -\frac{1}{\rho}\nabla p + \nu\nabla^2\mathbf{u} + \mathbf{f}, \tag{2}$$

where $\mathbf{u}$ represents the velocity field of the fluid, $p$ is the fluid pressure, $\rho$ is the fluid density, $\nu$ is the kinematic viscosity, and $\mathbf{f}$ is the body force per unit mass acting on the fluid. One-dimensional models are then derived by integrating these equations over the vessel cross-section to reduce the equations to a single spatial variable [23]. These models are defined in a three-dimensional space parameterized by an axial variable ($z$), where the functions $p(z,t)$, $q(z,t)$, and $A(z,t)$ represent the pressure, flow rate, and vessel lumen area at position $z$ and time $t$. The equations governing blood fluid dynamics under the assumption of Poiseuille flow are given by:

$$\frac{\partial A}{\partial t} + \frac{\partial q}{\partial z} = 0, \tag{3}$$

$$\frac{\partial q}{\partial t} + \frac{\partial}{\partial z}\left(\frac{4}{3}\frac{q^2}{A}\right) = -8\pi\nu\frac{q}{A} + \nu\frac{\partial^2 q}{\partial z^2} - \frac{A}{\rho}\frac{\partial p}{\partial z}, \tag{4}$$

where the kinematic viscosity of blood $\nu$ is conventionally set to $\nu = 3.77 \times 10^{-2}$ s cm$^{-2}$. To accurately predict the vessel response to blood flow, these equations are supplemented with a constitutive model. One-dimensional models offer a simplified yet valuable approach to studying blood fluid dynamics. However, the validity of these models may be compromised when they are employed on anatomies that contain numerous junctions or are affected by pathological conditions [7].

### B. Graph-Based Models for Cardiovascular Applications

Among the different data-driven methods available in the literature for addressing this problem [8]–[11], some recent advances have focused on using GNNs, as described in [7]. This method is based on MeshGraphNets, a framework for learning mesh-based simulations using GNNs [17]. Meshes are widely utilized in the numerical modeling of physical systems as they discretize complex geometries into interconnected elements, such as nodes and edges, facilitating the application of numerical methods like finite element analysis. The approach described in [17] is particularly effective for solving systems governed by partial differential equations, which describe a broad range of physical phenomena, including fluid dynamics, aerodynamics, and structural mechanics. MeshGraphNets leverage the inherent flexibility of mesh-based representations by encoding the state of the system into a graph, where nodes correspond to discrete points within the mesh and edges represent their spatial and functional relationships. Through message-passing operations between nodes, MeshGraphNets approximate the differential operators necessary to capture the underlying dynamics of the system. Additionally, the framework allows for adaptive mesh refinement, enabling the resolution of different regions of the system at varying levels of detail, thus improving both computational efficiency and scalability in simulating complex physical processes. In [7], the MeshGraphNet framework is used to process complex cardiovascular geometries represented as graphs, which are constructed by mapping the geometry of the cardiovascular system, where nodes and edges represent anatomical features like vessel segments and junctions. In this MeshGraphNet-based method [7], GNNs serve as a data-driven one-dimensional ROM. The architecture of the model incorporates a rollout phase, illustrated in Fig. 1 (b), where the network accepts as input the system state $\Theta^k$ and calculates an update facilitating the progression of the system state from $\Theta^k$ to $\Theta^{k+1}$. The state of the system includes pressure and flow rate at a particular time step and other relevant features detailed in Section III-B (such as cross-sectional area and parameters governing the boundary conditions). The application of the GNN is iterative: at the initial time step $t^0$, a predetermined initial condition is given as input to the network, while at each successive time step $t^k$ for $k > 0$, the previously estimated system state is provided. The GNN forward step is based on Multilayer Perceptrons (MLPs) with hidden layers numbered as $n_h$. The neuron count per hidden layer is uniform and equal to $n_s$. Each layer employs the LeakyReLU activation function. Furthermore, the output layer undergoes layer normalization.

For multi-input scenarios, the MLP inputs are concatenated into one tensor. As shown in Fig. 1, MeshGraphNet operates in three pivotal stages that define the forward step:

1) **Encode:** Node and edge attributes are encoded into latent features via MLPs. For each node feature $\mathbf{v}_i^k$, the latent representation is computed as $\mathbf{v}_i^{(0)} = f_{en}(\mathbf{v}_i^k) \in \mathbb{R}^{n_l}$, with the MLP $f_{en}$ mapping node features to an $n_l$-dimensional latent space. Edge features $\mathbf{w}_{ij}$ are analogously encoded to $\mathbf{w}_{ij}^{(0)} = f_{ee}(\mathbf{w}_{ij}) \in \mathbb{R}^{n_l}$.

2) **Process:** Conducted over $L$ iterations, this phase involves updating edge features initially, and then node features via aggregation functions, utilizing the MLPs $f_{pe}$ and $f_{pn}$:

$$\mathbf{w}_{ij}^{(l)} = f_{pe}^{(l)}\left(\mathbf{w}_{ij}^{(l-1)}, \mathbf{v}_i^{k,(l-1)}, \mathbf{v}_j^{k,(l-1)}\right) \in \mathbb{R}^{n_l}, \qquad (5)$$

$$\mathbf{v}_j^{k,(l)} = f_{pn}^{(l)}\left(\mathbf{v}_j^{k,(l-1)}, \sum_{i:\exists e_{ij}} \mathbf{w}_{ij}^{(l)}, \mathbf{w}_{in,j}, \mathbf{w}_{out,j}\right) \in \mathbb{R}^{n_l}. \qquad (6)$$

3) **Decode:** Latent node features are reverted to the output space using the MLP $f_{dn}$, and the output is a vector comprising the updated values of pressure and flow rate $[\delta p_i^k, \delta q_i^k] = f_{dn}(\mathbf{v}_i^{k,(L)}) \in \mathbb{R}^2$.

Upon finalization of the forward step, nodal pressure $p_i^k$ and flow rate $q_i^k$ values are updated to $p_i^{k+1}$ and $q_i^{k+1}$ by incorporating the respective deltas $\delta p_i^k$ and $\delta q_i^k$. The vector $\delta \mathbf{v}_i^k = [\delta p_i^k, \delta q_i^k, 0, \ldots, 0] \in \mathbb{R}^{17}$, and the function $\Psi_m$ are introduced, representing the cumulative effect of $m$ GNN applications during the rollout phase. Specifically,

$$\Psi_1(\Theta^k) = \bigcup_{i=1}^{N}\{\mathbf{v}_i^k + \delta\mathbf{v}_i^k\} \cup \bigcup_{i,j:\exists e_{ij}} \{\mathbf{w}_{ij}\}, \qquad (7)$$

$$\Psi_m(\Theta^k) = (\underbrace{\Psi_1 \circ \cdots \circ \Psi_1}_{m \text{ times}})(\Theta^k). \qquad (8)$$

The approximation of $p$ and $q$ at node $i$, after $m$ applications of the GNN, is denoted as $\Psi_m(\Theta^k)\mid_{p,i}$ and $\Psi_m(\Theta^k)\mid_{q,i}$, respectively.

### III. METHODS

As a baseline, we considered the MeshGraphNet-based architecture presented in [7]. To ensure a fair comparison, we kept a consistent graph representation, maintained the same data generation pipeline, and used the dataset considered in their study. These aspects are detailed within our work, allowing for a comprehensive understanding of the methodology and results.

### A. Notation

This work examines a collection of $G$ cardiovascular geometries, denoted by the set $\{\Omega_1, \Omega_2, \ldots, \Omega_G\}$, where each $\Omega_g$ represents the vascular structure of a specific patient. For each geometry, a directed graph is constructed with nodes

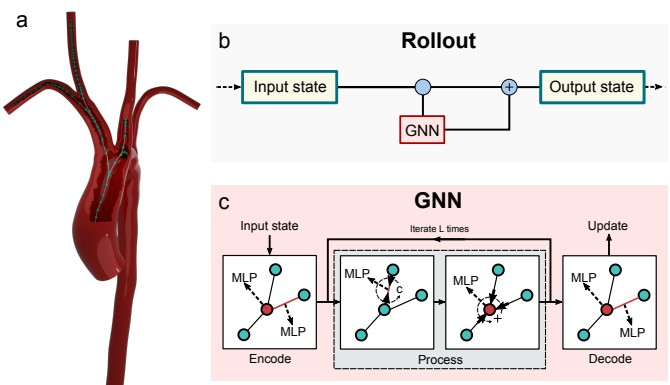

Fig. 1. Schematics of MeshGraphNet. (a) Aorta representation. The graph and relative node and edge features are generated in correspondence with the centerline of the vessels. (b) The rollout phase of the method is presented. Here, the initial state of the system $\Theta^k$ is fed into the GNN, which then generates the prediction for the update of the state variables. This update is merged with the current state to calculate $\Theta^{k+1}$. (c) The steps within the GNN are outlined. The node and edge attributes are initially encoded using an MLP, then processed $L$ times through aggregation functions, and finally decoded into the output domain.

$n_1^g, n_2^g, \ldots, n_{N^g}^g$ positioned along the central line of the vessel. The directed edge from node $i$ to node $j$ is denoted by $e_{ij}^g$.

The duration of a complete cardiac cycle is represented by $T_{cc}^g$. A temporal sequence starting from zero is considered, represented by $t^{0,g}, t^{1,g}, \ldots, t^{M,g}$, where $t^{0,g} = 0$ and $t^{M,g} = T_{cc}^g$, with a uniform time interval $\Delta t^g$ such that $t^{1,g} - t^{0,g} = t^{M,g} - t^{M-1,g}$. In the original [7], a unique $\Delta t^g$ is applied to all patients. Instead, in this work the model is adapted to variable cardiac cycle periods, including the addition of $\Delta t$ among the node features: this enhancement contributes to an increased generalizability of the model. At any given time $t^{k,g}$, the state of the system is characterized by $\Theta^{k,g}(\boldsymbol{\mu})$, where $\boldsymbol{\mu}$ denotes the system parameters, particularly related to the boundary conditions. A sequence of states $\Theta^{k,g}(\tilde{\boldsymbol{\mu}})$ for a specific set of parameters $\tilde{\boldsymbol{\mu}}$ is referred to as a trajectory. While these parameters and quantities are patient-specific, the superscript $g$ will be generally omitted in subsequent sections, unless otherwise stated.

### B. Dataset

We examine eight patient-specific geometries sourced from the Vascular Model Repository (VMR)[1]: Geometries 1-5 represent healthy aortas, Geometry 6 is an aortofemoral model featuring an aneurysm, Geometry 7 illustrates a healthy pulmonary system, and Geometry 8 is an aorta affected by coarctation. The last three geometries are selected because they exhibit features that have been identified in previous research as being particularly challenging for conventional physics-based models, such as aneurysm in Geometry 6, numerous junctions in Geometry 7, and stenosis in Geometry 8 [6]. Background information about these subjects, which is

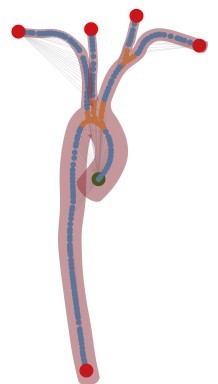

Fig. 2. Graph of a healthy aorta, depicting branch nodes in blue, junction nodes in orange, inlet node in green, and outlet nodes in red. Additionally, this figure highlights the boundary edges: edges connecting the model inlet or outlets to interior nodes.

necessary to understand their conditions fully, can be found in Table I.

TABLE I
INFORMATION OF THE PATIENTS AND THEIR ASSOCIATED GEOMETRY IDENTIFIERS IN THE VMR.

| # | Geometry | Age | Sex | Imaging | Anatomy | Condition |
|---|----------|-----|-----|---------|---------|-----------|
| 1 | 0090_0001 | 13 | M | MRI | Aorta | Healthy |
| 2 | 0091_0001 | 6 | M | CT | Aorta | Healthy |
| 3 | 0093_0001 | 11 | F | MRI | Aorta | Healthy |
| 4 | 0094_0001 | 23 | F | MRI | Aorta | Healthy |
| 5 | 0095_0001 | 26 | M | MRI | Aorta | Healthy |
| 6 | 0140_2001 | 76 | M | CT | Abd. Aorta | Aneurysm |
| 7 | 0080_0001 | 43 | F | MR | Pulmonary | Healthy |
| 8 | 0104_0001 | 11 | F | MRI | Aorta | Coarctation |

The selection of node and edge features relies on a thorough understanding of the problem. It is acknowledged that, under Poiseuille flow conditions, there exists a linear relationship between the flow rate $Q$ and the pressure drop $\Delta P$ in a vessel that approximates a cylindrical shape. The relationship can be expressed as:

$$\Delta P = RQ = \frac{8\mu L}{\pi r^4}Q. \tag{9}$$

The constant of proportionality $R$ denotes the resistance, which is dependent on the viscosity of the blood $\mu$, the length of the vessel $L$, and its radius $r$. In this analysis, blood viscosity and density are treated as constants, with $\mu = 0.04\,\mathrm{g\ cm^{-1}s^{-1}}$ and $\rho = 1.06\,\mathrm{g\ cm^{-3}}$, respectively, for all patients. These are not included as graph features since they are the same for all patients. Following fluid dynamics principles, the cross-sectional area is incorporated into the node features.

**Node features**. Average cross-sectional pressure $p_i^k \in \mathbb{R}^+$ and flow rate $q_i^k \in \mathbb{R}$ are considered at each centerline node $n_i$ as key indicators of the state of the system $\Theta^k$ at any given time $t^k$. These metrics are derived from planar cross-sections perpendicular to the centerline and within the encompassing cylindrical approximation of the vessel structure. The cross-sectional area $A_i \in \mathbb{R}$, representing the area of the vessel

lumen, is the area of the section passing through node $i$ and is considered a node feature. To distinguish different types of nodes within the graph, namely branch nodes, junction nodes, and inlet/outlet nodes (see Fig. 2), a one-hot encoding vector $\boldsymbol{\alpha}_i \in \mathbb{R}^4$ is employed. This is particularly crucial as the cross-sectional area can significantly vary at junctions, affecting hemodynamic behavior. Node feature vectors also encompass the tangential vector to the centerline, computed at node $n_i$, denoted as $\boldsymbol{\phi}_i \in \mathbb{R}^3$. This is essential for capturing the directional flow characteristics of the vessel over time. The entire cardiac cycle $T_{cc} \in \mathbb{R}^+$, the minimum pressure $p_{\min} \in \mathbb{R}^+$ and maximum pressure $p_{\max} \in \mathbb{R}^+$, and three parameters for the boundary conditions (RCR or resistance), namely $R_{i,p} \in \mathbb{R}^+$, $C_i \in \mathbb{R}^+$, and $R_{i,d} \in \mathbb{R}^+$, are also accounted for. Although this research relies on values $p_{\min}$ and $p_{\max}$ based on prior simulations, it is important to note that these values are typically known for the patient at hand and correspond to diastolic and systolic pressure in clinical practice. Furthermore, differently from [7], our work introduces the time step size $\Delta t \in \mathbb{R}^+$ as a node feature. This addition enhances the flexibility of the model, allowing it to adapt to varying cardiac cycle durations across different patients. By incorporating $\Delta t$ as a feature, the model can more accurately represent patient-specific physiological conditions, thereby achieving a higher degree of personalization. In summary, the feature vector for each node $n_i$ reads:

$$\mathbf{v}_i^k = \left[ p_i^k, q_i^k, A_i, \boldsymbol{\alpha}_i^T, \boldsymbol{\phi}_i^T, T_{cc}, \Delta t, p_{\min}, p_{\max}, R_{i,p}, C_i, R_{i,d} \right]^T \in \mathbb{R}^{17}. \quad (10)$$

**Edge features**. Three edge features are defined:

1) The difference between the position of nodes $n_i$ and $n_j$, i.e. $\mathbf{d}_{ij} = \mathbf{x}_j - \mathbf{x}_i \in \mathbb{R}^3$
2) The length of the shortest path connecting nodes $n_i$ and $n_j$, that is $z_{ij} \in \mathbb{R}^+$.
3) Firstly, it is necessary to introduce the so-called *boundary edges*, namely edges connecting boundary nodes to interior ones, in addition to physical edges. Therefore, a one-hot vector $\boldsymbol{\beta}_{ij} \in \mathbb{R}^4$ is introduced to represent the edge type: branch-to-branch, junction-to-junction, or boundary-to-interior (model inlet or outlets to interior). For simplicity, connections between branch and junction nodes share the same type as those within branches. Notably, only edges defining the centerline of the anatomical geometry (within branches and junctions) are considered when calculating shortest paths ($z_{ij}$). Although introducing boundary edges alters the graph structure, the different edge types preserve information about the original topology.

The combined edge features for the edge $e_{ij}$ are represented as a vector:

$$\mathbf{w}_{ij} = \left[ \frac{\mathbf{d}_{ij}^T}{\|\mathbf{d}_{ij}\|_2}, z_{ij}, \boldsymbol{\beta}_{ij}^T \right]^T \in \mathbb{R}^8. \quad (11)$$

Note that all features (with the exception of the unit vectors $\mathbf{t}_i$ and $\frac{\mathbf{d}_{ij}^T}{\|\mathbf{d}_{ij}\|_2}$) are normalized to conform to a standard Gaussian distribution $\mathcal{N}(0,1)$ using statistics derived from the dataset.

### C. LSTM architecture

LSTM networks [19] were designed to address the vanishing gradient problem in recurrent neural networks, particularly for tasks involving long-term dependencies. LSTMs are characterized by their gating mechanisms: these include input, forget, and output gates that manage information flow, enabling the network to retain and manipulate data over extended time periods. Graph LSTM can be used to enhance the efficiency of propagating long-term information across the graph structure [22]. Given two generic nodes $j$ and $z$, each of the LSTM units in this work contains input and output gates $i_j$ and $o_j$, a memory cell $c_j$, a hidden state $h_j$, the modulated input $u_j$, and a forget gate $f_{jz}$ for each edge, allowing node $j$ to aggregate information from its neighbors accordingly. The equations for the proposed Graph LSTM are:

$$i_j^k = \sigma \left( W^i \mathbf{v}_j^k + \sum_{z \in N_j} U^i h_z^{k-1} + b^i \right), \quad (12)$$

$$f_{jz}^k = \sigma \left( W^f \mathbf{w}_{jz}^k + U^f h_z^{k-1} + b^f \right), \quad (13)$$

$$o_j^k = \sigma \left( W^o \mathbf{v}_j^k + \sum_{z \in N_j} U^o h_z^{k-1} + b^o \right), \quad (14)$$

$$u_j^k = \tanh \left( W^u \mathbf{v}_j^k + \sum_{z \in N_j} U^u h_z^{k-1} + b^u \right), \quad (15)$$

$$c_j^k = i_j^k \odot u_j^k + \sum_{z \in N_j} f_{jz}^k \odot \sigma \left( c_j^{k-1} \right), \quad (16)$$

$$h_j^k = o_j^k \odot \tanh \left( c_j^k \right). \quad (17)$$

Where $\mathbf{v}_j^k$ represents the node feature of node $j$ at timestep $k$, $\mathbf{w}_{jz}^k$ is the edge feature at timestep $k$ of the edge connecting nodes $j$ and $z$, $\odot$ denotes the Hadamard product, $\sigma(\cdot)$ is the sigmoid activation function, and $z \in N_j$ denotes the neighbor node of $j$. The gradients of parameters $W$, $U$, and $b$ are computed from the loss introduced in (20), and are updated according to the computed gradient following the backpropagation algorithm. Differently from [7], where training was conducted on small consecutive time steps, in this work, we train on the entire temporal sequence. This approach is necessary since the output of each LSTM cell serves as the input for the next. Specifically, the quantity $h_z^{k-1}$, computed at the previous time step $k-1$, is needed. The dependence of the computation of each cell on the previous one increases memory requirements since the computational graph becomes larger and larger as the number of steps increases. For the development of the proposed model, the LSTM structure presented in [22] is used as a starting point. However, the following modifications have been implemented:

1) Following (16) as in the original reference [22], wherein the activation function was not applied to $c_j$, resulted in divergence towards infinity. The application of the sigmoid activation function ensures that the output of (16)

remains within a bounded range, addressing potential problems with uncontrolled growth.

2) In the original LSTM unit, matrices $U$ were dependent on the edge type, represented as $U_{m(j,z)}$. This configuration made the model computationally intensive, exceeding the resources detailed in Section IV-B. Consequently, we used edge-type-independent matrices $U$ to prevent the model from becoming prohibitively expensive from the computational viewpoint.

3) In the initial cell configuration, the formulation of $f_{jz}^k$ is the following:

$$f_{jz}^k = \sigma\left(W^f \mathbf{v}_j^k + U_{m(j,z)}^f h_z^{k-1} + b^f\right). \tag{18}$$

The quantity $\mathbf{v}_j$ (node features) was modified to $\mathbf{w}_{jz}$ (edge features) in order to incorporate information regarding edge types. This adjustment was made in response to the transition of matrices to edge-type independence, as described in Step 2.

We embedded the LSTM architecture in the MeshGraphNet-based method [7] *forward step*, outlined in Section II-B. As represented in Fig. 3, we implemented the LSTM unit between the *process step* and *decode step*. However, we made a minor adjustment in our proposed framework: we do not encode edge features due to the excessive computational resource requirements relative to our available capacity, delineated in Section IV-B.

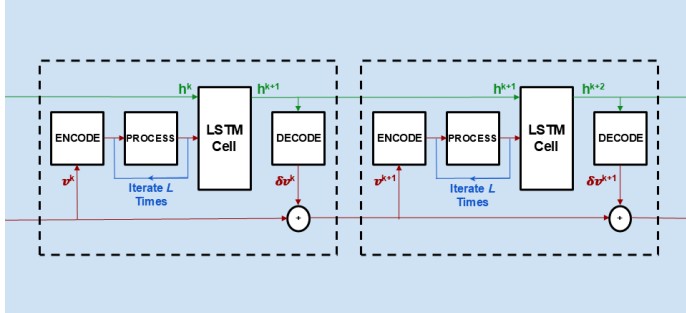

Fig. 3. The proposed GNN-LSTM architecture. At time $k$, the node features $\mathbf{v}^k$ are encoded using an MLP. Edge features $\mathbf{w}$ and encoded node features $f_{en}(\mathbf{v}^k)$ are then processed $L$ times using aggregation operations and sent as input in the LSTM cell. Node features are then decoded into the output space. The cumulative effect $\delta \mathbf{v}_i^k$ of the GNN is used to update the pressure and flow rate at time $k + 1$.

### D. Training and Evaluation

Let us define the training dataset, $\{\Omega_g\}_{g \in \mathcal{T}}$, where $\mathcal{T}$ is a subset of the indices ranging from 1 to $G$. Unlike the previous approaches [7] that trained on small strides of consecutive time steps, the proposed method involves training on the entire sequence to capture the precise values of nodal pressure $\hat{p}_i^{k,g}$ and flow rate $\hat{q}_i^{k,g}$, as explained in Section III-C. The single-geometry Mean Squared Error (MSE) is defined as follows:

$$\text{MSE}^g = \frac{1}{N^g} \sum_{l=1}^{M^g} \sum_{i=1}^{N^g} b_i \left[(\hat{p}_i^{l,g} - \Psi_l(\Theta^{0,g})|_{p,i})^2 + (\hat{q}_i^{l,g} - \Psi_l(\Theta^{0,g})|_{q,i})^2\right],$$
$$\tag{19}$$

where $b_i = c$ for boundary nodes and $b_i = 1$ otherwise ($c$ is a hyperparameter), and $\Psi_l$ denotes the result of applying the LSTM $l$ consecutive times. Consequently, the loss function $\mathcal{L}$ for the GNN is given by:

$$\mathcal{L} = \sum_{g \in \mathcal{T}} \frac{\text{MSE}^g}{|\mathcal{T}|}. \tag{20}$$

Stochastic gradient descent, coupled with the Adam optimizer, is employed to optimize the loss function. Each trajectory within the dataset is considered as a single data point for training purposes. For validation, $k$-fold cross-validation (with $k = 10$) is used and the partitioning of the data into training and testing sets is configured such that a proportion of $1 - \frac{1}{k}$ of the trajectories is allocated to the training set, while $\frac{1}{k}$ is allocated to the testing set. The dataset is partitioned to prevent data leaks, ensuring that the augmented data for each simulation is assigned to the same set (train or test) as the original simulation. In order to evaluate the performance of the model, the mean rollout error is used, both for pressure and flow rate, obtained on the test dataset post-training. Specifically, for a patient labeled as $g$, the error metrics for pressure $e_p^g$ and flow rate $e_q^g$ were computed according to the following expressions:

$$e_p^g = \frac{\sum_{i \in B^g} \sum_{k=1}^{M^g} (\hat{p}_i^{k,g} - \Psi_k(\Theta^{0,g}|_{p,i}))^2}{\sum_{i \in B^g} \sum_{k=1}^{M^g} (\hat{p}_i^{k,g})^2}, \tag{21}$$

$$e_q^g = \frac{\sum_{i \in B^g} \sum_{k=1}^{M^g} (\hat{q}_i^{k,g} - \Psi_k(\Theta^{0,g}|_{q,i}))^2}{\sum_{i \in B^g} \sum_{k=1}^{M^g} (\hat{q}_i^{k,g})^2}, \tag{22}$$

where $\hat{p}_i^{k,g}$ and $\hat{q}_i^{k,g}$ denote the exact measurements of pressure and flow rate at time $t^k$ and at node $n_i$ for patient $g$. It is important to note that these error calculations only included node indices within the branched regions, denoted as $B^g$. The term *Average Error* refers to the following metric:

$$\text{Average Error} = \frac{1}{2}(e_p^g + e_q^g). \tag{23}$$

### IV. EXPERIMENTAL DESIGN

#### A. Hyperparameter optimization

In this study, hyperparameters were optimized by using Tune [24], a Python library for hyperparameter tuning. The monitored objective function is the Average Error (23) defined in Section III-D. With respect to the healthy Geometries 1-5, we reach the optimal parameters reported in Table II after 50 iterations.

#### B. Experimental Setup

Central to the proposed method was the use of the Deep Graph Library (DGL) [25], a Python library designed for processing and manipulating GNNs. The Deep Graph Library is complemented with Modulus[2], an open-source deep learning framework for building, training, and fine-tuning deep learning

---

[2]https://developer.nvidia.com/modulus

TABLE II
LIST OF TUNED HYPERPARAMETERS.

| Hyperparameter | Value |
|---|---|
| Learning rate | $7,6 \cdot 10^{-4}$ |
| Learning rate decay | $1,3 \cdot 10^{-3}$ |
| Training Batch size | 17 |
| Weight of boundary nodes ($c$ in 19) | 1 |
| Number of hidden layers | 42 |
| GNN latent size | 46 |
| MLP latent size | 182 |
| MLP Number of hidden layers | 1 |
| Process iterations | 1 |

models using state-of-the-art physics-informed machine learning methods, which has been used to develop the GPU version of the code. Our method is implemented in PyTorch [26], a Python library that allows for the design of deep learning models. Hyperparameter tuning, Training, and inference run over Expanse, a supercomputer that is collaboratively designed by Dell and the San Diego Supercomputer Center (SDSC). Expanse GPU nodes were employed, each featuring four NVIDIA V100s (32 GB SMX2) connected via NVLINK, supplemented with dual 20-core Intel Xeon 6248 CPUs.

## V. RESULTS

The proposed method is compared with the MeshGraphNet-based method [7]. To ensure a fair comparison, we take the following steps:

- Hyperparameter Optimization was performed for both models,
- The $\Delta t$ feature was added to the MeshGraphNet-based method as well,
- The same computational resources for training as described in Subection IV-B were used,
- The same dataset outlined in Section III-B was employed.

Fig. 4 shows the advantages achieved by the proposed method in comparison to the MeshGraphNet-based one across three different performance indicators:

1) **Average Error**: The proposed method shows a significant increase in accuracy with an average error of $1.39\%$, which is a $65\%$ relative improvement over the $4\%$ error rate of the MeshGraphNet-based method for Geometries 1 to 5, and a $2.46\%$ error rate with a $52\%$ improvement over the $5.14\%$ error rate for Geometries 6 to 8.

2) **GPU Training Efficiency**: When it comes to training efficiency on a GPU, the proposed method completes an epoch in 16.41 seconds on average for Geometries 1 to 5, which is $57\%$ faster than the MeshGraphNet-based method that takes 38.41 seconds, and in 19.65 seconds for Geometries 6 to 8, a $73\%$ improvement over the 73.3 seconds of the MeshGraphNet-based method.

3) **CPU Training Efficiency**: Similarly, CPU training times per epoch are optimized. The proposed method requires on average 172 seconds per epoch for Geometries 1 to 5, which is a $59\%$ improvement in comparison to the 418 seconds required by the MeshGraphNet-based

method, and 475 seconds for Geometries 6 to 8, a $60\%$ improvement over the 1190 seconds.

The observed improvement in training performance can be attributed to the methodology employed in the training process. The MeshGraphNet-based method uses overlapping strides of time steps, which means the same samples are observed multiple times during training. This redundancy increases computational effort without necessarily improving learning efficiency. In contrast, the proposed model embeds time steps more efficiently by avoiding this overlap, thereby utilizing the data more effectively and reducing unnecessary computations.

**Cardiac Cycle Period Flexibility**. Additionally, the proposed model introduces the ability to select different periods for the cardiac cycle of each patient, providing better adaptability and potential for customization, as opposed to the fixed period in the MeshGraphNet-based method. This flexibility could translate into better generalizability in diverse simulation scenarios.

| | Geometries | MeshGraphNet-based Model | Proposed Model | Relative Improvement |
|---|---|---|---|---|
| **Average Error** | 1 to 5 | 4% | 1.39% | **65%** |
| | 6 to 8 | 5.14% | 2.46% | **52%** |
| **GPU Training Time per epoch** | 1 to 5 | 38.41 s | 16.44 s | **57%** |
| | 6 to 8 | 73.3 s | 19.65 s | **73%** |
| **CPU Training Time per epoch** | 1 to 5 | 418 s | 172 s | **59%** |
| | 6 to 8 | 1190 s | 475 s | **60%** |

Fig. 4. Performance metrics for the MeshGraphNet-based method [7] and the proposed method, showcasing improvements in average error rates, and computational efficiency during training on both GPU and CPU.

## VI. CONCLUSIONS

In this study, a novel approach for cardiovascular simulations that integrates LSTM networks into a GNN framework has been successfully developed and validated. This method, which builds upon the MeshGraphNet-based framework [7], demonstrates considerable advancements in terms of accuracy, computational efficiency, and adaptability to varying cardiac cycles. The integration of LSTM networks enables the model to capture long-term dependencies effectively, an aspect crucial for accurately simulating cardiovascular dynamics over extended periods. Key results of this work include:

1) Significant improvement in accuracy: the average error reduction achieved by the proposed method, particularly in geometries with complex cardiovascular structures, underscores its superior predictive capability over the MeshGraphNet-based method. This improvement is critical for advancing the reliability and clinical relevance of cardiovascular simulations.

2) Enhanced computational efficiency: the notable reduction in training times on both GPU and CPU platforms highlights the efficiency of the proposed method, making it a more practical option for broader applications, including time-sensitive clinical settings.

3) Flexibility in cardiac cycle period: the ability of the proposed method to adapt to different cardiac cycle periods for individual patients represents a significant step towards personalized medicine. This flexibility enhances the applicability of the method to a wider range of patient-specific scenarios, thereby increasing its utility in personalized healthcare.

Future research directions include further validation of the method with larger and more diverse datasets, including pathological cases, to robustly assess its generalizability. A further avenue for exploration involves assessing the impact of modifying the feature set on the accuracy of the method. Specifically, the possibility of excluding boundary condition parameters while holding the patient-specific data typically employed to establish these conditions presents a notable potential improvement over existing methodologies. Currently, the calibration of boundary conditions in physics-based simulations is a pivotal process, often executed by adjusting parameters in simpler surrogate models (like zero- or one-dimensional ROMs) through Bayesian optimization. This process typically relies on functions aiming to evaluate the accuracy of the surrogate model in replicating crucial physiological metrics, such as systolic and diastolic pressures. By directly integrating these physiological metrics into the neural networks, there may be an opportunity to eliminate the need for boundary condition calibration, thereby streamlining the process from medical image acquisition to the evaluation of calibrated simulation results. Additionally, alternative methods, such as the application of transformers, should be further investigated. A challenge might include the need for more training data, which, in the field of scientific computing, may be computationally expensive to generate and not always readily available. Another challenge could be effectively dealing with topologically diverse graphs. Nevertheless, through continued exploration and refinement, valuable insights and improvements may be uncovered.

## ACKNOWLEDGMENT

This work was supported by the Ermenegildo Zegna Founder's Scholarship. The authors gratefully acknowledge the San Diego Supercomputer Center (SDSC) for providing the computational resources necessary for training the models presented in this paper.

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
