# OpenReview forum: "A Novel LSTM and Graph Neural Networks Approach for Cardiovascular Simulations"
_IEEE.org/EMBS/BHI/2024/Conference — IEEE BHI'24_

### Official Review · Reviewer_L51P · 2024-07-31
**A promising application of LSTM and GNN for cardiovascular simulations**

**Overall Rating:** 7
**Confidence:** 3

**Other Quality Metrics:**

a) Clarity of writing: Great

b) Clinical Significance: Good

c) Methodological Novelty: Good

d) Experiments and Results: Great

**Questions For The Authors:**

1. Would it be possible to run the simulations on an average gaming desktop computer that has a dedicated NVIDIA graphics card or is it required to have a supercomputer?
a. If having a supercomputer is a requirement, would it be possible for the authors to discuss the options for reproducing this study and/or building on it by other researchers as well as generalizability of the model proposed to be used by wider audiences such as hospitals and clinics.
b. If having a supercomputer is a requirement, who will be able to utilize the model presented?
c. Authors are encouraged to demonstrate the performance of the model on other lesser powerful hardware such as an average gaming desktop computer.

2. It appears that even though there are 269 models in the Vascular Model Repository (VMR), authors used only 8 of them. Could authors elaborate more on why such a small number of the models were chosen except the fact that these three were found to be challenging in the past study? Wouldn’t it be better if authors utilized a larger portion of the dataset to build a more generalized model?
a. Authors are suggested to use a larger dataset to demonstrate the generalization capabilities of the proposed model.

**Strengths:**

1. The paper has a solid mathematical background.
2. Overall structure, flow and the narrative of the paper is good.
3. Included figures, tables, and the equations help readers to understand the paper better.
4. Sufficient number of recent papers in the field of interest were cited.
5. Drawn conclusions are adequate.

**Summary Of The Paper:**

In this study, a novel method which combines long short-term memory (LSTM) networks with graph neural networks (GNNs) to be used for cardiovascular simulations is proposed. By making use of the key concepts from fluid dynamics that is applied to blood flow, and their previous study, authors were able to improve their previous model up to 65%.

**Weaknesses:**

1.The paper heavily relies on reference [25] to explain some of the key concepts of the study as well as to highlight the impact of it. Authors are suggested to incorporate more from the original study into this paper to reduce the amount of dependency.

2. Abstract needs to be improved as in its current state, it is hard to understand the key concepts of the study from just by reading the abstract. It focuses too much on the result and a comparison with a previous study but the details are too vague.
a. For example, authors stated that this study builds upon a previous one that uses a MeshGraphNet-based framework, but the nature of this framework and the previous work is unclear.
b. It is unclear why reduced-order models of cardiovascular simulations are needed in the first place just by reading the abstract.
It is unclear what the current methods are lacking (motivation of the study)
c. The term "cardiovascular personalization" is unclear, making it hard to understand the impact of the paper.

3. References should be numbered in the same order as they appear in the text. Authors are kindly asked to renumber the papers they cited taking this rule into consideration. For example, [42] comes after [5], but should have been followed by [6].

4. A few typographical and grammatical errors exist. Authors are recommended to proofread the entire paper and address any typographical and grammatical mistakes.

5. Fonts used in the figures are way too small, making figures illegible unless the document is zoomed-in. Authors are suggested to reconsider the figures in terms of readability and improve it without breaking the cohesion of the paper.

6. In the Introduction, the sentence “Zero-dimensional models have proven to be useful in numerous studies; see [6]–[10].” does not really add anything to the paper without reading the 5 papers given. Similarly, the sentence “Refer to [11]–[24] for examples of uses of such models in cardiovascular simulations.” is not a good way to relay information.
a. Authors are recommended to either omit these references or summarize them in the text considering that the paper has 47 references which is quite high and unusual for an 8-page conference paper.

7. It would be better if the authors included a brief comparison with the other existing methods to better highlight the impact of the paper.

---

### Official Review · Reviewer_CQF9 · 2024-08-07
**The paper introduces a method for personalizing clinical options, but it needs more validation with medical experts and a better dataset. The approach is analytically strong but lacks details on personalization and subject background.**

**Overall Rating:** 7
**Confidence:** 3

**Other Quality Metrics:**

(a) Clarity of writing:  Good
(b) Clinical Significance: Good
(c) Methodological Novelty: Great
(d) Experiments and Results: Excellent

**Questions For The Authors:**

1. How specifically is affected by the ROM mentioned in the study?
2. During the literature review, what other approaches were considered? Why is the performance compared only with the MeshGraphNet model?
3. What challenges were encountered during the experimentation other than mentioned in this paper?

**Strengths:**

1. The mathematical calculations in the paper are clearly presented and accurate.
2. The machine learning methods used are effective, and the approach is innovative.
3. The paper evaluates its results using the MeshGraphNet model, showing a focused approach.

**Summary Of The Paper:**

The paper presents a method for customizing clinical options for users. However, the study's outcomes are within the range of what's acceptable for medical AI but lack a substantial quality dataset necessary for reliable results. Further validation is needed with input from medical practitioners to ensure the approach is accurate and feasible. The paper demonstrates a strong analytical approach, resulting in good accuracy.

**Weaknesses:**

1. The paper mentions flexibility but doesn't explain how personalization is achieved.
2. It references data from 8 subjects without providing background information about these subjects, which is necessary to understand their conditions.
3. The validation relies mainly on engineering and machine learning methods, but clinical AI should also involve input from medical experts.

---

### Official Review · Reviewer_ZUu4 · 2024-08-18
**Interesting paper with significant improvement over previous methods**

**Overall Rating:** 8
**Confidence:** 4

**Other Quality Metrics:**

(a) Clarity of writing: Excellent
(b) Clinical Significance: Great
(c) Methodological Novelty: Great
(d) Experiments and Results: Good

**Questions For The Authors:**

- I don't have any questions to the authors

**Strengths:**

- The paper is well organized into different sections.
- All experiments and results are clearly described.
- Appropriate references are present in this paper.
- The novel method described in this paper results in markedly better performance and faster training time.
- This work has the potential to be clinically significant.

**Summary Of The Paper:**

The authors develop a method to personalize cardiac cycle analysis for different patients by incorporating LSTM networks with GNNs. This helps in better performance of the proposed model in comparison to previously developed methods, as well as helps in making the process of training this model computationally cheaper.

**Weaknesses:**

- Plot of training loss curve would have been a useful addition to this paper.
- During k-fold cross-validation, the value of k is missing.

---

### Decision · Program_Chairs · 2024-09-23

Accept